# Neuroevolution for Parameter Adaptation in Differential Evolution

Vladimir Stanovov *, Shakhnaz Akhmedova and Eugene Semenkin

Institute of Informatics and Telecommunication, Reshetnev Siberian State University of Science and Technology, 660037 Krasnoyarsk, Russia; shahnaz@inbox.ru (S.A.); eugenesemenkin@yandex.ru (E.S.)
* Correspondence: vladimirstanovov@yandex.ru

**Abstract:** Parameter adaptation is one of the key research fields in the area of evolutionary computation. In this study, the application of neuroevolution of augmented topologies to design efficient parameter adaptation techniques for differential evolution is considered. The artificial neural networks in this study are used for setting the scaling factor and crossover rate values based on the available information about the algorithm performance and previous successful values. The training is performed on a set of benchmark problems, and the testing and comparison is performed on several different benchmarks to evaluate the generalizing ability of the approach. The neuroevolution is enhanced with lexicase selection to handle the noisy fitness landscape of the benchmarking results. The experimental results show that it is possible to design efficient parameter adaptation techniques comparable to state-of-the-art methods, although such an automatic search for heuristics requires significant computational effort. The automatically designed solutions can be further analyzed to extract valuable knowledge about parameter adaptation.

**Keywords:** differential evolution; neuroevolution; parameter adaptation; neuroevolution of augmented topologies





## 1. Introduction

The area of Computational Intelligence (CI), which includes Artificial Neural Networks (ANN), Fuzzy Logic Systems (FLS) and Evolutionary Algorithms (EA), is one of the most rapidly developing directions nowadays. These algorithms find applications in many fields where prediction [1], modelling [2,3] or optimization problems [4] are solved. Evolutionary algorithms are mostly applied for solving optimization problems, and they are often state-of-the-art approaches in many areas, such as multiobjective optimization, constrained optimization and black-box numerical optimization [5]. The algorithms which are developed for solving single-objective numerical optimization problems often serve as a basis for other techniques and are applicable to a wide range of real-world problems [6].

Among the numerical optimization techniques developed in the area of evolutionary computation [7], Differential Evolution (DE) is currently one of the most widely used approaches. The original DE algorithm proposed in [8] appeared to be efficient and relatively simple in terms of implementation [9]. Further development of DE, which mainly included new mutation strategies and parameter adaptation techniques, made it one of the most efficient optimization methods today [10]. However, despite a certain level of success, the topic of parameter adaptation remains one of the most studied, as DE is highly sensitive to parameter values.

As opposed to the classical approach, whereby new algorithms are developed by hand and based on the expert knowledge of researchers, several studies have made attempts to automate the process of searching for optimization algorithms and techniques. In particular, some studies addressed the problem of designing new algorithms directly [11,12] with advanced genetic programming techniques, such as PushGP. Other studies, for example, Refs. [13,14] focused on creating certain search operators for the existing algorithms. All these methods could be generalized as Hyper-Heuristic (HH) approaches [15], which are

applied to the Automated Design of Algorithms (ADA) problem [16]. Such methods are of particular interest as they are not bounded by human perception of the problem and corresponding bias in algorithmic design.

In this study, the problem of designing a parameter adaptation technique for differential evolution is considered. In particular, the Neuroevolution of Augmented Topologies (NEAT) [17] algorithm is applied to set the scaling factor and crossover rate values in DE. The Artificial Neural Networks (ANN) designed by NEAT are trained directly on the DE algorithm, implementing the offline parameter adaptation approach [18], when parameter values are chosen based on the set of experiments during training [19]. The training is performed on a set of benchmark problems introduced for the Congress on Evolutionary Computation (CEC) competition on bound-constrained numerical optimization in 2022 [20], and the testing is performed on the CEC 2021 competition benchmark [21]. This study is the continuation of our previous research, namely [22], where Genetic Programming (GP) [23] was applied to the same problem, but a with different benchmark set. The main features of the current study are:

1. Both the scaling factor $F$ and crossover rate $Cr$ are controlled by a single network designed by NEAT;
2. $\epsilon$-lexicase selection is applied, where the cases are the results of each run on each benchmark function;
3. Difference-based mutation is applied in NEAT for enhanced weights tuning;
4. Evaluation of the designed artificial neural networks involves both convergence speed and the final obtained function value.

The performed experiments show that the automatically designed parameter adaptation techniques in the form of neural networks are able to deliver comparable level of performance not only on the set of functions on which they were trained but also on different benchmarks.

The rest of the paper is organized as follows. The next section contains a short description of DE and NEAT, as well as the proposed approach. After this, the experimental setup and results are given, including the discussion. Finally, the conclusions and directions of further work are provided.

## 2. Materials and Methods

### 2.1. Differential Evolution

Differential Evolution is a population-based evolutionary algorithm proposed by Storn and Price [24]. The key idea of DE is the usage of difference vectors between individuals, which allows for efficient exploitation of the functions' landscape. DE starts with an initialization step, whereby points are randomly generated within the boundaries $[xmin_j, xmax_j]$: $x_{i,j}, i = 1 \ldots NP, j = 1 \ldots D$, where $D$ is the problem dimension and $NP$ is the population size.

The main loop of DE consists of mutation, crossover and selection operators. The mutation is the main part of DE, which makes it different from other algorithms. Most recent DE-based algorithms use the current-to-pbest/1 strategy [25], which generates mutant (donor) vector $v_i$ for each target vector $x_i$ as follows:

$$v_{i,j} = x_{i,j} + F(x_{pbest,j} - x_{i,j}) + F(x_{r1,j} - x_{r2,j}), \tag{1}$$

where *pbest* is an index of one of the $p * 100\%$ best individuals, $r1$ and $r2$ are randomly chosen indices from $[1, NP]$. $F$ is called the scaling factor and is usually chosen in the range $[0, 1]$. Note that *pbest*, $r1$ and $r2$ are generated to be different from each other.

After mutation, the generated donor vector $v_i$ is combined with a target vector $x_i$ to produce a trial vector $u_i$ with crossover. The commonly used crossover operation is

binomial crossover, in which the trial vector receives randomly chosen components from the mutant vector with a probability $Cr \in [0, 1]$:

$$u_{i,j} = \begin{cases} v_{i,j}, & \text{if } rand(0,1) < Cr \text{ or } j = jrand \\ x_{i,j}, & \text{otherwise} \end{cases}, \quad (2)$$

where $jrand$ is a randomly chosen index from $[1, D]$, required to make sure that the trial vector is smaller than the target vector to avoid unnecessary fitness calculations.

Before calculating the goal function value for the trial vector $u_i$ the Bound Constraint Handling Method (BCHM) should be applied. One of the often-used approaches is called midpoint target [26]:

$$u_{i,j} = \begin{cases} \frac{xmin_j + x_{i,j}}{2}, & \text{if } u_{i,j} < xmin_j \\ \frac{xmax_j + x_{i,j}}{2}, & \text{if } u_{i,j} > xmax_j \end{cases}. \quad (3)$$

After applying BCHM and the fitness calculation, the selection step is performed. If the goal function value of the trial vector is small than that of the target vector (minimization problem), then the replacement occurs:

$$x_{i,j} = \begin{cases} u_{i,j}, & \text{if } f(u_i) \leq f(x_i) \\ x_{i,j}, & \text{if } f(u_i) > f(x_i) \end{cases}. \quad (4)$$

Although the general scheme of DE is relatively simple, it comes with the price of high sensitivity to parameter values. Nowadays there are many studies addressing this issue [27], and further several well-known approaches will be considered. In [28] the jDE algorithm was proposed, in which the parameter values were adapted as follows:

$$F_{i,t+1} = \begin{cases} random(F_l, F_u), & \text{if } random(0,1) < \tau_1 \\ F_{i,t}, & \text{otherwise} \end{cases}, \quad (5)$$

$$CR_{i,t+1} = \begin{cases} random(0,1), & \text{if } random(0,1) < \tau_2 \\ CR_{i,t}, & \text{otherwise} \end{cases}. \quad (6)$$

where $F_l$ and $F_u$ are the lower and upper boundaries for $F$, and $\tau_1$ and $\tau_2$ control the frequency of $F$ anc $Cr$ changes, usually set to 0.1. The jDE algorithm updates $F$ and $Cr$ if the trial vector is better than the target vector storing the successful values. The jDE approach is known to be quite efficient, and its modifications jDE100 [29] and j2020 [30] demonstrated high efficiency on bound-constrained test problems.

Another branch of parameter adaptation techniques started with the JADE algorithm [31], which ,together with the current-to-pbest/1 mutation strategy, proposed automatic parameter tuning. The crossover rate $Cr$ and scaling factor $F$ are generated for each crossover and mutation using normal distribution with the mean set to $\mu_{Cr}$, standard deviation set to 0.1, and Cauchy distribution with location parameter set to $\mu_F$ and scale parameter 0.1. As the sampled values may fall away from the $[0, 1]$ interval, they were sampled again until they were inside the boundaries. However, if the sampled $F > 1$, then it was set to $F = 1$. The JADE algorithm also included updating the memory values $\mu_{Cr}$ and $\mu_F$ depending on the successful values.

Further development of JADE resulted in the SHADE (Success-History Adaptive Differential Evolution) algorithm, originally proposed in [32]. SHADE further developed the ideas of JADE by introducing $H$ memory cells, each containing a pair of $(M_{F,h}, M_{Cr,h})$ values, which are used for parameters sampling as follows:

$$\begin{cases} F = randc(M_{F,k}, 0.1) \\ Cr = randn(M_{Cr,k}, 0.1) \end{cases}, \quad (7)$$

where *randc* is a Cauchy distributed random value, *randn* is a normally distributed random number and $k$ is chosen from $[1, H]$ for generating each trial vector.

At the end of the generation, the successful values of the $F$ and $Cr$ parameters were stored in $S_F$ and $S_{Cr}$, as well as the corresponding goal function improvements $S_{\Delta f}$, where $\Delta f_i = |f(u_i) - f(x_i)|$. The memory cells were updated one by one using the values calculated with weighted Lehmer mean [33]:

$$mean_{wL} = \frac{\sum_{j=1}^{|S|} w_j S_j^2}{\sum_{j=1}^{|S|} w_j S_j},$$ (8)

where $w_j = \frac{S_{\Delta f_j}}{\sum_{k=1}^{|S|} S_{\Delta f_k}}$, $S$ is either $S_{Cr}$ or $S_F$. The Lehmer mean results are further used to set the new $M_F$ and $M_{Cr}$ values at iteration $t$ for one of the memory cells $h$:

$$\begin{cases} M_{F,h}^{t+1} = 0.5(M_{F,h}^t + mean_{(wL,F)}) \\ M_{CR,h}^{t+1} = 0.5(M_{Cr,h}^t + mean_{(wL,Cr)}) \end{cases},$$ (9)

The index of the updated memory cell $h$ is incremented every generation and is set to 1 once it reaches the number of cells $H$.

The JADE and SHADE algorithms also maintain an external archive of inferior solutions, which is composed of parent vectors replaced during selection. Initially, the archive $A$ is empty and is filled with solutions until it reaches its predefined maximum size $NA$. After this, the archive is updated by replacing a randomly chosen individual in the archive with a new one. The archived solutions are used in the current-to-pbest/1 mutation strategy in the $r2$ index: It is chosen from either the population or the archive. Thus, the archive set allows the diversity of generated trial vectors to be improved. The SHADE algorithm has gained high popularity due to its high efficiency, and has become a basis for a variety of new approaches [34–37].

After SHADE, the L-SHADE algorithm was developed [38], which included a control technique for the population size $NP$, one of the three main parameters of DE. The Linear Population Size Reduction (LPSR) method decreases the population size based on current computational resource and removes the worst individuals from the population. The population size is recalculated every generation as follows:

$$NP_{g+1} = round(\frac{NP_{min} - NP_{max}}{NFE_{max}} NFE + NP_{max}),$$ (10)

where $NP_{min} = 4$ and $NP_{max}$ are the minimal and initial population sizes, and $NFE$ and $NFE_{max}$ are the current and maximal number of function evaluations. Later the modification of the LPSR method was proposed, in which the population size decreases non-linearly (NLPSR) [39]. This approach is inspired by the one applied in the AGSK algorithm [40] and updates the population size as follows:

$$N_{g+1} = round((N_{min} - N_{max})NFE_r^{1-NFE_r} + N_{max}),$$ (11)

where $NFE_r = \frac{NFE}{NFE_{max}}$ is the ratio of the current number of fitness evaluations. The NLPSR approach sets the population size to smaller values compared to the LPSR.

Although the mentioned parameter adaptation techniques are considered to be quite efficient, some studies show that there is room for further improvement [41], for example, with biased parameter adaptation [42]. The automated search for efficient adaptation heuristics is one of the possible ways of receiving valuable knowledge about differential evolution [22]. In the next subsection, the NEAT approach will be briefly described.

### 2.2. Neuroevolution

The problem of evolving artificial neural network topology is one of the most important areas of studies because the number of neurons and their connectivity significantly influences the efficiency and complexity of the resulting solutions. Several attempts were made in combining ANN with evolutionary algorithms, and the Neuroevolution of Augmented Topologies (NEAT) [17] remains one of the most promising directions of studies. The main advantage of NEAT is that it allows both the topology and the weights of an ANN to be evolved thanks to specific solution representation and operators. This allows NEAT to be efficient in solving a variety of problems, such as classification, regression and control, among others [43].

The main features of the NEAT algorithm are the usage of historical markers (or innovation numbers), speciation mechanism and initialization with minimal structures. NEAT starts with a population of simple networks, where only inputs and outputs are connected, and the weights are assigned randomly. The search process consists of adding new nodes and connections, i.e., augmenting the ANN topology via mutation and crossover operators. The encoding scheme in NEAT is based on a set of nodes and a set of connections, with each element in these sets having several properties. In particular, the nodes have historical markers, a type of node (input, output, hidden) and an activation function, while connections have historical markers, source and destination nodes numbers, a weight value and an activation flag.

The historical markers indicate the moment when a certain node or connection was added. These markers are required to align the corresponding genes during crossover, which helps to solve the competing conventions problem, which occurs when the encoding scheme allows several ways of expressing the same solution [17].

The NEAT algorithm follows a general scheme, starting with initialization, followed by selection, crossover, mutation and speciation. The selection mechanism could be based on any of the mechanisms widely used in EC, such as rank-based or tournament selection. The crossover step combines the genetic information of two parents by aligning the genes having identical innovation numbers. The offspring is composed of the genes, which are randomly chosen from either the first or second parent. As the parents may have different chromosome lengths, there is a possibility that disjoint genes or excess genes will occur. The disjoint genes are the ones that are present in one parent, but not in the other, while there is still a common part after that. The excess genes are the tail genes of the parent individuals, which are present only in one of the parents, and there is no common part after that. The disjoint and excess genes are taken from the parent with higher fitness during crossover.

The original NEAT algorithm proposed two mutation schemes: adding a new connection and adding a node to the connection. When adding a connection, a pair of nodes is randomly selected, and a new connection gene is created at the end of the chromosome with a new innovation number. The weight of the newly generated connection is randomly sampled. When adding a node to the connection, one of the existing connections is randomly chosen and split into two, and a randomly generated hidden node is placed in between. One of the weights of the two new connections is set to 1, and the other keeps the previous value. The new historical markers are assigned to both new connections and the node.

The speciation step implements the innovation protection role in NEAT. When the new connections or nodes are added to the individual, there is a significant chance that they will decrease the efficiency of the solution, but further development of this solution may result in more promising ones. To avoid deleting new solutions and still keep the population evolving, the speciation (or niching) procedure is applied, which uses innovation numbers to determine similarities and differences between individuals and create subgroups within population. In NEAT, the first individual creates the first species, and the subsequent individuals are either added to this species or create their own based on compatibility or similarity distance. If this distance is smaller than a certain threshold, then an individual belongs to the species. The distance combines the information about the number of excess

genes $E$, the number of disjoint genes $D$ and the average difference in weights of the matching genes $\overline{W}$ as follows:

$$\delta = c_1 \cdot E + c_2 \cdot D + c_3 \cdot \overline{W}, \tag{12}$$

where $c_1$, $c_2$ and $c_3$ are the importance coefficients. At the end of each generation, the offspring individuals are assigned to species. After this, the best representatives of each species create the new generation. The general scheme of NEAT is shown in Algorithm 1.

---

**Algorithm 1** NEAT

---

1: **for** $gen = 1$ to $NG$ **do**
2:     Initialize population $P$ with minimal solutions, calculate fitness $fit_i$, $i = 1, \ldots, N$
3:     Assign species numbers to every individual
4:     **for** $i = 1$ to $N$ **do**
5:         Perform tournament-based selection ($t = 2$) to get index $tr$
6:         Perform crossover between $P_i$ and $P_{tr}$, save offspring to $O_i$
7:         Select the mutation operator to be used
8:         Perform mutation on $O_i$ and calculate fitness
9:     **end for**
10:     Perform speciation of combined parents $P$ and offspring $O$ populations
11:     Create new population from the representatives of every species
12: **end for**
13: Return best found solution

---

A variety of studies have investigated ways of improving NEAT and applying it to different fields, including studies on fitness landscape [44], using different activation functions [45], meta-analysis [46] and using it as hyper-heuristic [47]. In the next subsection, the proposed approach for evolving parameter adaptation heuristics with NEAT is described.

*2.3. Proposed Approach*

The problem of tuning parameters is one of the most studied in evolutionary computation [48]. The idea of automating the search for parameter adaptation heuristics is, to the best of our knowledge, relatively new, and was considered in [22]. In [22], genetic programming (GP) [49] was applied to find heuristics for controlling the $F$ and $Cr$ values via symbolic regression [50,51], and separate solutions were evolved for $F$ and for $Cr$. In this paper, instead of GP, which was previously used for designing hyper-heuristics [52], the NEAT approach is applied. One of the advantages of NEAT over GP is that it is capable of evolving structures with several inputs and outputs, and allows interaction between inputs and outputs within one solution, unlike GP, where two separate trees were built to control both $F$ and $Cr$ at the same time.

As a baseline approach to be used for hyper-heuristic design with NEAT, a simplified version of the recently developed NL-SHADE-RSP approach [39] was considered. Being the winner of the CEC 2021 competition on bound-constrained single-objective optimization for biased, shifted and rotated cases, NL-SHADE-RSP represents a state-of-the-art approach. Thus, improving its performance further is a challenging task. However, the version of NL-SHADE-RSP used in this study is simplified (NL-SHADE-RSP$_s$), i.e., some of the specific parameter adaptation techniques are removed in order to allow improved flexibility for NEAT during heuristics search.

NL-SHADE-RSP$_s$ uses non-linear population size reduction, as described above, and the current-to-pbest/r mutation strategy with rank-based selection. The additional selection mechanism, proposed in [53], and further studied in [54], assigns ranks $R_i = e^{-i/NP}$ in an array sorted by fitness values, with largest ranks assigned to better individuals. These

ranks are further used to calculate the probabilities of choosing an individual from the population for the $r2$ index:

$$pr_i = \frac{R_i}{\sum_{j=1}^{NP} R_j}, \tag{13}$$

where $i = 1 \ldots NP$. Hence, better individuals with larger fitness have higher chances of being used in the mutation. The $pb$ parameter in the current-to-pbest/r mutation strategy is gradually increased, i.e., the initial value of $pb_{min}$ is set to 0.2, and the number of individuals to choose *pbest* from is linearly increased as follows:

$$pbest = max(2, NP(0.2 + 0.1\frac{NFE}{NFE_{max}})). \tag{14}$$

The minimal value of $pb$ is set to 2 individuals.

NL-SHADE-RSP$_s$ uses only binomial crossover and does not apply any heuristics for controlling the $Cr$ value depending on current computational resource. NL-SHADE-RSP$_s$ does not include adaptive archive probabilities and crossover rate sorting. The pseudocode of NL-SHADE-RSP$_s$ is presented in Algorithm 2.

The inputs for the NEAT individuals included the following parameters:

- The ratio of the currently used computational resource $NFE_r = \frac{NFE}{NFE_{max}}$;
- The success ratio, defined as $SR = \frac{|A|}{NP}$;
- The individual number ratio $IN = \frac{i}{NP}$;
- The last good $F$ value for individual $i$;
- The last good $Cr$ value for individual $i$.

For the third parameter, $IN$, to be informative, the whole population was sorted by fitness at the beginning of each generation. It allows different values of $F$ and $Cr$ to be assigned depending on the quality of an individual. This makes it possible to replicate the sorting of $Cr$ values, which was applied in NL-SHADE-RSP, by setting the appropriate weights in the NEAT individual. The DE with built-in NEAT parameter adaptation does not use memory cells, and the resulting values returned by the designed neural network are used to sample $F$ and $Cr$ with Cauchy and normal distribution instead of memory cell values.

The NEAT algorithm applied in this study had several features, which were specifically added for the problem of parameter adaptation heuristic design. First of all, Difference-Based Mutation (DBM), proposed in [55], was added, as it allows better fine-tuning of the weight coefficients. In addition, according to the conclusions of [45], the following activation functions were used:

- Linear: equals $z$;
- Negative: $-z$;
- Absolute value: $|z|$;
- Squared: $z^2$;
- Unsigned step function: equals 1 if $z > 0$;
- Sigmoid: $\frac{1}{1+e^{-z}}$;
- ReLU: $max(0, z)$;
- Gaussian: $e^{\frac{-z^2}{2}}$;
- Hyperbolic tangent: $tanh(z)$;
- Sine: $sin(\pi z)$;
- Cosine: $cos(\pi z)$.

Here, $z$ is the sum of inputs to a certain node. The default function is linear, and the mutation may choose any of these functions with equal probability.

---

**Algorithm 2** NL-SHADE-RSP$_s$

---

1: Set $NP_{max} = 23D$, $NP = NP_{max} D$, $NFE_{max}$,
2: $H = 20D$, $A = \emptyset$, $M_{F,r} = 0.5$, $M_{Cr,r} = 0.9$, $k = 1$
3: $NA = NP$, $p_A = 0$, $g = 0$
4: Initialize population $P_0 = (x_{1,j}, \ldots, x_{NP,j})$ randomly
5: **while** $NFE < NFE_{max}$ **do**
6:　　$S_F = \emptyset$, $S_{Cr} = \emptyset$, $n_A = 0$
7:　　Sort and rank population according to fitness $f(x_i)$
8:　　**for** $i = 1$ to $NP$ **do**
9:　　　　Current memory index $r = randInt[1, H + 1]$
10:　　　Crossover rates $Cr_i = randn(M_{Cr,r}, 0.1)$
11:　　　$Cr_i = min(1, max(0, Cr))$
12:　　　**repeat**
13:　　　　　$F_i = randc(M_{F,r}, 0.1)$
14:　　　**until** $F_i \geq 0$
15:　　　$F_i = min(1, F_i)$
16:　　**end for**
17:　　**for** $i = 1$ to $NP$ **do**
18:　　　**repeat**
19:　　　　　$pbest = randInt(1, NP * p)$
20:　　　　　$r1 = randInt(1, NP)$
21:　　　　　**if** $rand[0, 1] < p_A$ **then**
22:　　　　　　　$r2 = randInt(1, NP)$
23:　　　　　**else**
24:　　　　　　　$r2 = randInt(1, NA)$
25:　　　　　**end if**
26:　　　**until** $i \neq pbest \neq r1 \neq r2$
27:　　　**for** j=1 to D **do**
28:　　　　　$v_{i,j} = x_{i,j} + F(x_{pbest,j} - x_{i,j}) + F(x_{r1,j} - x_{r2,j})$
29:　　　**end for**
30:　　　Binomial crossover with $Cr$
31:　　　Calculate $f(u_i)$
32:　　　**if** $f(u_i) < f(x_i)$ **then**
33:　　　　　$x_i \rightarrow A_{randInt[1,NA]}$, $x_i = u_i$
34:　　　　　$F \rightarrow S_F$, $Cr \rightarrow S_{Cr}$, $\Delta f_i = f(x_i) - f(u_i)$
35:　　　**end if**
36:　　**end for**
37:　　Get $NP_{g+1}$ and $NA_{g+1}$ with NLPSR
38:　　**if** $|A| > NA_{g+1}$ **then**
39:　　　Remove random individuals from the archive
40:　　**end if**
41:　　**if** $NP_g > NP_{g+1}$ **then**
42:　　　Remove worst individuals from the population
43:　　**end if**
44:　　Update $M_{F,k}$, $M_{Cr,k}$
45:　　$p_A = |A|/NP$
46:　　$k = mod(k, H) + 1$, $g = g + 1$
47: **end while**
48: Return best solution $x_{best}$

---

The mutation step also included two additional mutation operations other than adding weight and adding a connection:

- Mutating random node: The type of operation performed in a randomly chosen node is changed to another one, and the new innovation number is assigned to the node.
- Assigning random weights: Every connection is mutated with a probability of $1/NC$, and $NC$ is the number of connections. The connection receives either weight chosen

from $randn(0, 0.1)$ or $randn(1, 0.1)$. Otherwise, the current value of the weight is used as a mean value to generate new ones as follows: $w = randn(w, 0.01)$, where $w$ is the current weight value. With a probability of $1/NC$, each weight is either activated or deactivated.

- Difference-based mutation, which utilizes the idea of difference vectors used in DE to mutate new weights. For this purpose, three individuals are selected: the target one with index $i$ and two others with randomly chosen indices $r1$ and $r2$. The genes with the same innovation numbers are identified, and if they have different weights, they are marked as mutating. Next, the following equations are applied: $w_{i,j} = w_{i,j} + F_{NEAT} * (w_{r1,j} - w_{r2,j})$, where $j$ is the index of matching genes, and $F_{NEAT} = 0.5$.

In the NEAT algorithm used in this study, several mutation operations could be applied to the same individual. The probabilities of mutation were set as shown in Table 1, in accordance with the values used in [55].

**Table 1.** Probabilities of mutation operators in NEAT.

| Mutation Type | Probability to Use |
|---|---|
| Add connection | 0.1 |
| Adding node to connection | 0.3 |
| Mutating random node | 0.2 |
| Assigning random weights | 0.2 |
| Difference-based mutation | 0.1 |

The crossover operation was performed only for the weights, i.e., disjoint and excess genes were not added to the offspring, and each matching gene in the offspring received a weight value from one of the parents with equal probabilities. After the crossover and mutation steps, the newly generated offspring is checked for cycles in the network graph. If cycles are present, then the offspring is discarded and generated again.

The evaluation of NEAT solutions was performed on a set of benchmark functions from the CEC 2022 competition on numerical optimization. The benchmark consists of 12 functions defined for 10-dimensional and 20-dimensional search spaces. To evaluate a single parameter adaptation heuristic designed by NEAT, 10 independent runs were performed for each of the 12 test functions in $10D$ case. On each of the runs, the result consisted of the best reached function value and the computational resource spent to reach the goal, if it was reached. The CEC 2022 competition rules introduced the ranking of solutions based on both convergence speed and the best-found solution, where, between the two algorithms which found the solution, the one that did it faster was ranked higher. In order to take such a mechanism into account, the following evaluation metric was proposed:

$$EM_{fn,r} = \begin{cases} f_{r,NFE_{max}}, & \text{if } f_{r,NFE_{max}} > 10^{-8} \\ \frac{NFE_s}{NFE_{max}} \cdot 10^{-9}, & \text{otherwise} \end{cases} \tag{15}$$

where $NFE_s$ is the number of goal function evaluations where the $10^{-8}$ threshold was reached, $f_{r,NFE_{max}}$ is the best function value at the end of the search on run $r$ and $fn$ is the function number. Such an evaluation metric allows two criteria to be combined into a single one when those runs in which the algorithm found the global optimum are better (have smaller $EM_r$) than those runs in which the optimum was not found.

The quality of each parameter adaptation heuristic designed by NEAT is described by a matrix of 12 by 10 values, which should be properly utilized in the search process. The classical selection mechanisms, such as tournament selection, are not able to efficiently compare such evaluation metrics, so for this purpose, lexicase selection was applied [56]. Lexicase selection implements a parent selection mechanism, which allows population diversity to be increased by giving a chance to be selected to each individual capable of

solving some part of the problem. Lexicase selection was shown to improve the problem-solving power in program synthesis [57], as well as classifier systems [58] and other approaches [59]. As the evaluation metric values in the current study are real numbers, $\epsilon$-lexicase selection was applied [60] because the standard case passing criteria is too strict for continuous problems.

Lexicase selection works by adding the population into the selection pool, shuffling the fitness values (*EM* matrix in this case, flattened to a vector $EMF_t$, $t = 1, \ldots, 120$), removing the cases with fitness values larger than the fitness of the first case + $\epsilon_1$ level, and if more than one individual remains, then the first case is removed and the algorithm is repeated. If there are no more fitness cases, or only one case in the pool, then the parent individual is chosen from the pool. The $\epsilon_t$ level is determined for a case $t$ as follows:

$$\epsilon_t = median_j(EMF_{t,j} - mean(EMF_t)). \tag{16}$$

The idea of applying $\epsilon$-lexicase selection in this study was to allow NEAT to evolve specialists solutions are capable of solving certain types of optimization problems, so that later such individuals could be combined to obtain universal parameter adaptation heuristics.

Another mechanism in addition to lexicase selection applied in this study is the behavior-based speciation inspired by [61]. The idea of this modified speciation mechanism is to add the performance metrics on difference cases to the distance measure used in speciation:

$$\delta = c_1 \cdot E + c_2 \cdot D + c_3 \cdot \overline{W} + c_4 \cdot BD, \tag{17}$$

$$BD = \sum_{t=1}^{120} (1 - \frac{1}{1 + |EMF_{j,t} - EMF_{k,t}|}), \tag{18}$$

where importance coefficients are set to $c_1 = c_2 = c_3 = 1$ and $c_4 = 10$, and $j$ and $k$ are the indices of compared individuals.

The speciation mechanism is designed to maintain a fixed number of species (7) during the algorithm's run. For this purpose, the speciation threshold is adaptively adjusted every generation. The representative of each species is chosen as the best individual in this species, and other individuals are compared to the representative. At the end of each generation, all individuals are assigned to species sorted by fitness values, and only several best individuals of each species are copied to the next generation. The fitness values for each individual $fit_i$, used in speciation only, are determined using the Friedman ranking procedure of the results on all functions. For this, the *EMF* values of all individuals on each run and function are sorted and ranked, and the fitness values are set as the sum of ranks. This fitness assignment mechanism was applied in [22].

The results of benchmarking a certain parameter adaptation heuristic designed by NEAT are random, as NL-SHADE-RSP$_s$ intensively uses random values during the search. To avoid promoting solutions which were occasionally ranked very high, all individuals are re-evaluated at the beginning of each generation. This means that some solutions will be evaluated more than once. However, it allows heuristics with genuinely poor performance to be filtered out. The described neuroevolutionary algorithm will be further referred to as HA-NEAT-DBM (Heterogeneous Activation NEAT with Difference-Based Mutation).

In the next section, the experimental setup and results are provided.

## 3. Results

The computational experiments were performed to evaluate the ability of the proposed HA-NEAT-DBM to design parameter adaptation heuristics. For this purpose, the algorithm was trained on CEC 2022 benchmark functions and then evaluated. Such an approach implements offline parameter adaptation [19,62], as opposed to online adaptation [63], where parameters are tuned during the optimization process.

Training a single ANN with NEAT requires significant computational effort, and in the case of the current study, it is multiplied by the goal function complexity, which requires 10 runs of the NL-SHADE-RSP$_s$ algorithm on 12 test functions from the CEC 2022 competition [20]. Therefore, to reduce calculation time, HA-NEAT-DBM and NL-SHADE-RSP$_s$ were both implemented in C++11, compiled with GCC and ran on the OpenMPI-powered cluster of seven AMD Ryzen 1700 processors with eight cores each. There were 50 independent runs of HA-NEAT-DBM performed, each on a separate core, the computational resource was set to 10000 HA-NEAT-DBM evaluations and the time required to perform the training was around 16 days running 24/7. Testing the algorithms, however, required much less computation, as only the best solutions of each run were analyzed. The best solution was determined by the Friedman ranking of the results at the last generation, as described in the previous section. The post-processing of results, statistical tests and graphs were performed with Python 3.8 and matplotlib. The main parameters of NL-SHADE-RSP$_s$ are given in Algorithm 2.

The first set of efficiency comparison experiments was performed on the CEC 2022 benchmark functions. The designed parameter adaptation heuristics, presented in the form of networks, were compared with the baseline NL-SHADE-RSP$_s$ and the original NL-SHADE-RSP, which is the winner of the previous year's competition CEC 2021 for biased, shifted and rotated functions. To compare the efficiency, all the ANNs were tested on all 12 functions, 30 independent runs and $10D$ and $20D$ cases. For every function, the Mann–Whitney statistical test was performed, with significance level $p = 0.01$, normal approximation and tie-breaking, i.e., runs with equal results were assigned averaged ranks, and the standard score $Z$ value was calculated. In Table 2, the aggregated results are shown, i.e., the number of wins (+), ties (=) and losses ($-$) of one of the 50 heuristic compared to NL-SHADE-RSP$_s$ or NL-SHADE-RSP.

The comparison of statistical tests results in Table 2 demonstrates that the neuroevolutionary parameter adaptation performs better than the standard SHADE parameter adaptation of the simplified NL-SHADE-RSP$_s$ algorithm in both $10D$ and $20D$ scenarios. It should be mentioned, however, that the heuristics are usually better on the same functions: For example, in the $20D$ case, almost all automatically designed parameter adaptation techniques are better on 3 functions and comparable on 9. These significant improvements were observed for F1 (Shifted and full Rotated Zakharov Function), F3 (Shifted and full Rotated Expanded Schaffer's f6 Function) and F5 (Shifted and full Rotated Levy Function), and the improvements were in terms of the number of function evaluations required to reach the optimum. On all other functions, the results were equal or comparable. As for the $10D$ scenario, the designed heuristics were better for most of the functions, except F4 (Shifted and full Rotated Non-Continuous Rastrigin's Function) and F8 (Hybrid Function 3, including Katsuura, HappyCat, Griewank's plus Rosenbrock's, Modified Schwefel's and Ackley's).

The comparison with the original NL-SHADE-RSP, which uses specific crossover rate adaptation, adaptive archive probabilities and exponential crossover, shows that the designed heuristics for parameter adaptation are still capable of competing with this approach, but the results are worse in general. In the $10D$ case, there are from 2 to 4 losses, although the number of wins is larger, and in the $20D$ case, there are almost always 3 wins and 1 loss. This means that these algorithms behave differently on different functions, and the designed parameter adaptation techniques have other principles compared to success-history adaptation. In particular, losses on $10D$ functions were observed on F6 (Hybrid Function 1, including Bent Cigar, HGBat and Rastrigin's functions), F8, F10 (Composition Function 2, including Rotated Schwefel's, Rotated Rastrigin's and HGBat functions) and F12 (Composition Function 4, including HGBat, Rastrigin's, Modified Schwefel's, Bent Cigar, High Conditioned Elliptic and Expanded Schaffer's functions). Here, the best parameter adaptation technique was designed at run 29: It performs well against both NL-SHADE-RSP$_s$ and NL-SHADE-RSP.

The results in Table 2 seem promising, but it should be taken into account that the HA-NEAT-DBM algorithm used these functions to train and select best solutions here. Thus, a certain overfitting may have occurred here. The results in the 20$D$ case could be considered as testing, but even though the dimension of the problems is different, the functions have the same structure and similar landscape. To test the efficiency of parameter adaptation the previous year's benchmark set, CEC 2021 was considered. In CEC 2021, there are eight different benchmarks, which combine different function alternations, such as bias, shift and rotation [21]. Table 3 contains the aggregated comparison on all eight benchmarks. Note that in the case of CEC 2021, the statistical tests did not use information about convergence speed, according to the competition rules.

The numbers in Table 3 are the number of wins, ties and losses over all 8 benchmarks, each containing 10 test functions. Analyzing the results of CEC 2021, the difference between different automatically discovered heuristics could be clearly seen. For example, in some cases, the heuristics are always equal or worse than the NL-SHADE-RSP$_s$ parameter adaptation, and in other cases, the heuristics are usually better. For example, runs 2, 22, 28, 36, 42 and 49 could be considered as relatively successful as they have comparable or better performance on both 10$D$ and 20$D$. However, when compared to the original NL-SHADE-RSP, the results of the ANNs designed by HA-NEAT-DBM are relatively poor: Even run 49, which is one of the most successful, has only 13 wins on 10$D$ and 43 losses. Such a difference in performance is probably due to the fact that NL-SHADE-RSP was specifically designed for the CEC 2021 benchmark, and has special parameter adaptation mechanisms that allow it to reach high a performance level.

The experiments on the CEC 2021 benchmark also allow the efficiency of the designed heuristics to be determined in solving biased, shifted and rotated problems. To perform the comparison, the baseline NL-SHADE-RSP$_s$ was compared to the heuristic from run 49 on different benchmarks, and the results are shown in Table 4.

The results of statistical tests in Table 4 show that in the basic scenario, without bias, shift and rotation of functions, the heuristic from run 49 performs quite well in the 10$D$ and 20$D$ cases; however, if the functions are shifted, the results are worse. The rotation procedure, which significantly changes the landscape, does not influence the performance of the heuristic compared to NL-SHADE-RSP$_s$. Moreover, in the 20$D$ case, it appears to be better for the Rotation and Bias benchmarks on 4 functions out of 10. In particular, here the improvements were achieved on F2 (Shifted and Rotated Schwefel's function), F3 (Shifted and Rotated Lunacek bi-Rastrigin function), F6 (Hybrid Function 2, including Expanded Schaffer, HGBat, Rosenbrock's and Modified Schwefel's functions) and F7 (Hybrid Function 3, including Expanded Schaffer, HGBat, Rosenbrock's, Modified Schwefel's and High Conditioned Elliptic functions). Therefore, it can be concluded that the designed parameter adaptation allows rotated problems to be solved more efficiently.

For a deeper understanding of how the designed parameter adaptation techniques operate during the algorithm run, in Figure 1, the graphs of inputs to the neural network and outputs from it are shown for CEC 2022 test functions, and the solution from run 29 is considered.

**Table 2.** Comparison of HA-NEAT-DBM heuristics on CEC 2022 benchmark functions, Mann–Whitney tests.

| Run | NL-SHADE-RSP$_s$ | | NL-SHADE-RSP | |
| --- | --- | --- | --- | --- |
| | **10***D* | **20***D* | **10***D* | **20***D* |
| 1 | 5+/6=/1− | 3+/9=/0− | 6+/3=/3− | 3+/8=/1− |
| 2 | 6+/4=/2− | 3+/9=/0− | 8+/1=/3− | 3+/8=/1− |
| 3 | 7+/4=/1− | 3+/9=/0− | 7+/3=/2− | 3+/8=/1− |
| 4 | 6+/3=/3− | 3+/9=/0− | 7+/1=/4− | 3+/8=/1− |
| 5 | 4+/6=/2− | 2+/9=/1− | 6+/2=/4− | 2+/9=/1− |
| 6 | 6+/6=/0− | 3+/9=/0− | 7+/2=/3− | 3+/8=/1− |
| 7 | 5+/6=/1− | 3+/9=/0− | 6+/2=/4− | 3+/8=/1− |
| 8 | 5+/6=/1− | 3+/9=/0− | 6+/3=/3− | 3+/8=/1− |
| 9 | 6+/4=/2− | 3+/9=/0− | 7+/2=/3− | 3+/8=/1− |
| 10 | 6+/4=/2− | 3+/9=/0− | 7+/1=/4− | 3+/8=/1− |
| 11 | 5+/5=/2− | 3+/9=/0− | 6+/2=/4− | 3+/8=/1− |
| 12 | 7+/4=/1− | 3+/9=/0− | 7+/3=/2− | 3+/8=/1− |
| 13 | 5+/5=/2− | 3+/9=/0− | 7+/1=/4− | 3+/8=/1− |
| 14 | 7+/5=/0− | 3+/9=/0− | 7+/3=/2− | 3+/8=/1− |
| 15 | 7+/5=/0− | 3+/9=/0− | 7+/3=/2− | 3+/8=/1− |
| 16 | 8+/2=/2− | 3+/9=/0− | 7+/2=/3− | 3+/8=/1− |
| 17 | 5+/5=/2− | 3+/9=/0− | 6+/2=/4− | 3+/8=/1− |
| 18 | 6+/5=/1− | 3+/9=/0− | 7+/3=/2− | 3+/8=/1− |
| 19 | 7+/5=/0− | 3+/9=/0− | 7+/3=/2− | 3+/8=/1− |
| 20 | 6+/5=/1− | 3+/9=/0− | 7+/1=/4− | 3+/8=/1− |
| 21 | 6+/4=/2− | 3+/9=/0− | 6+/3=/3− | 3+/8=/1− |
| 22 | 6+/4=/2− | 3+/9=/0− | 7+/2=/3− | 3+/8=/1− |
| 23 | 5+/7=/0− | 3+/9=/0− | 7+/2=/3− | 3+/8=/1− |
| 24 | 5+/6=/1− | 3+/9=/0− | 6+/3=/3− | 3+/8=/1− |
| 25 | 6+/3=/3− | 3+/9=/0− | 7+/1=/4− | 3+/8=/1− |
| 26 | 6+/5=/1− | 3+/9=/0− | 7+/2=/3− | 3+/8=/1− |
| 27 | 6+/4=/2− | 3+/9=/0− | 7+/2=/3− | 3+/8=/1− |
| 28 | 6+/5=/1− | 3+/9=/0− | 8+/1=/3− | 3+/8=/1− |
| 29 | 8+/3=/1− | 3+/9=/0− | 8+/1=/3− | 3+/8=/1− |
| 30 | 7+/4=/1− | 3+/9=/0− | 7+/2=/3− | 3+/8=/1− |
| 31 | 7+/3=/2− | 3+/9=/0− | 7+/2=/3− | 3+/8=/1− |
| 32 | 6+/5=/1− | 3+/9=/0− | 7+/1=/4− | 3+/8=/1− |
| 33 | 5+/5=/2− | 3+/9=/0− | 6+/2=/4− | 3+/8=/1− |
| 34 | 6+/5=/1− | 3+/9=/0− | 7+/3=/2− | 3+/8=/1− |
| 35 | 6+/4=/2− | 3+/9=/0− | 7+/2=/3− | 3+/8=/1− |
| 36 | 6+/5=/1− | 3+/9=/0− | 7+/2=/3− | 3+/8=/1− |
| 37 | 6+/5=/1− | 2+/10=/0− | 6+/2=/4− | 2+/9=/1− |
| 38 | 6+/4=/2− | 2+/9=/1− | 7+/3=/2− | 3+/8=/1− |
| 39 | 7+/4=/1− | 3+/9=/0− | 7+/2=/3− | 3+/8=/1− |
| 40 | 5+/5=/2− | 3+/9=/0− | 7+/2=/3− | 3+/8=/1− |
| 41 | 6+/4=/2− | 3+/9=/0− | 7+/1=/4− | 3+/8=/1− |
| 42 | 6+/6=/0− | 3+/9=/0− | 7+/3=/2− | 3+/8=/1− |
| 43 | 3+/5=/4− | 2+/10=/0− | 6+/2=/4− | 2+/9=/1− |
| 44 | 6+/5=/1− | 3+/9=/0− | 7+/3=/2− | 3+/8=/1− |
| 45 | 6+/5=/1− | 3+/9=/0− | 7+/2=/3− | 3+/8=/1− |
| 46 | 6+/5=/1− | 3+/9=/0− | 7+/2=/3− | 3+/8=/1− |
| 47 | 6+/5=/1− | 3+/9=/0− | 7+/1=/4− | 3+/8=/1− |
| 48 | 5+/5=/2− | 2+/10=/0− | 6+/2=/4− | 2+/9=/1− |
| 49 | 7+/5=/0− | 3+/9=/0− | 7+/3=/2− | 3+/8=/1− |
| 50 | 6+/5=/1− | 3+/9=/0− | 7+/2=/3− | 3+/8=/1− |

**Table 3.** Comparison of HA-NEAT-DBM heuristics on CEC 2021 benchmark functions, Mann–Whitney tests.

| Run | NL-SHADE-RSP$_s$ | | NL-SHADE-RSP | |
| | $10D$ | $20D$ | $10D$ | $20D$ |
| --- | --- | --- | --- | --- |
| 1 | 7+/52=/21− | 9+/42=/29− | 9+/25=/46− | 7+/19=/54− |
| 2 | 13+/44=/23− | 20+/34=/26− | 7+/29=/44− | 7+/21=/52− |
| 3 | 0+/57=/23− | 6+/41=/33− | 6+/28=/46− | 8+/15=/57− |
| 4 | 1+/41=/38− | 4+/35=/41− | 5+/28=/47− | 6+/19=/55− |
| 5 | 0+/44=/36− | 1+/33=/46− | 6+/26=/48− | 5+/18=/57− |
| 6 | 7+/57=/16− | 14+/48=/18− | 9+/27=/44− | 8+/19=/53− |
| 7 | 0+/45=/35− | 2+/40=/38− | 7+/26=/47− | 6+/18=/56− |
| 8 | 14+/46=/20− | 15+/43=/22− | 8+/26=/46− | 9+/20=/51− |
| 9 | 1+/42=/37− | 3+/39=/38− | 5+/28=/47− | 6+/19=/55− |
| 10 | 0+/48=/32− | 6+/37=/37− | 6+/27=/47− | 8+/17=/55− |
| 11 | 2+/45=/33− | 3+/40=/37− | 8+/23=/49− | 8+/17=/55− |
| 12 | 3+/62=/15− | 13+/50=/17− | 10+/28=/42− | 8+/19=/53− |
| 13 | 2+/45=/33− | 2+/40=/38− | 6+/26=/48− | 7+/17=/56− |
| 14 | 3+/59=/18− | 6+/45=/29− | 7+/31=/42− | 8+/16=/56− |
| 15 | 2+/49=/29− | 6+/40=/34− | 7+/26=/47− | 8+/17=/55− |
| 16 | 1+/46=/33− | 5+/38=/37− | 4+/28=/48− | 8+/17=/55− |
| 17 | 2+/40=/38− | 3+/38=/39− | 7+/24=/49− | 6+/19=/55− |
| 18 | 1+/47=/32− | 5+/38=/37− | 6+/27=/47− | 7+/18=/55− |
| 19 | 10+/57=/13− | 19+/49=/12− | 10+/26=/44− | 8+/19=/53− |
| 20 | 3+/44=/33− | 3+/38=/39− | 8+/23=/49− | 6+/19=/55− |
| 21 | 1+/50=/29− | 5+/40=/35− | 8+/25=/47− | 8+/16=/56− |
| 22 | 15+/46=/19− | 20+/43=/17− | 9+/28=/43− | 8+/21=/51− |
| 23 | 0+/55=/25− | 6+/49=/25− | 8+/24=/48− | 7+/18=/55− |
| 24 | 4+/41=/35− | 4+/38=/38− | 9+/23=/48− | 6+/18=/56− |
| 25 | 0+/40=/40− | 3+/37=/40− | 5+/28=/47− | 7+/18=/55− |
| 26 | 0+/54=/26− | 7+/48=/25− | 11+/23=/46− | 8+/17=/55− |
| 27 | 0+/45=/35− | 3+/39=/38− | 6+/27=/47− | 7+/18=/55− |
| 28 | 8+/59=/13− | 21+/44=/15− | 7+/30=/43− | 7+/20=/53− |
| 29 | 0+/49=/31− | 4+/41=/35− | 9+/23=/48− | 8+/17=/55− |
| 30 | 0+/53=/27− | 7+/36=/37− | 9+/26=/45− | 8+/17=/55− |
| 31 | 0+/41=/39− | 3+/39=/38− | 5+/27=/48− | 6+/19=/55− |
| 32 | 2+/45=/33− | 3+/41=/36− | 9+/24=/47− | 7+/18=/55− |
| 33 | 2+/44=/34− | 2+/38=/40− | 8+/22=/50− | 6+/18=/56− |
| 34 | 4+/55=/21− | 6+/50=/24− | 12+/22=/46− | 7+/18=/55− |
| 35 | 13+/46=/21− | 16+/45=/19− | 10+/26=/44− | 6+/21=/53− |
| 36 | 10+/51=/19− | 21+/43=/16− | 6+/32=/42− | 6+/25=/49− |
| 37 | 3+/43=/34− | 2+/38=/40− | 9+/22=/49− | 7+/17=/56− |
| 38 | 4+/59=/17− | 7+/54=/19− | 12+/28=/40− | 7+/20=/53− |
| 39 | 0+/51=/29− | 6+/41=/33− | 6+/27=/47− | 8+/17=/55− |
| 40 | 2+/43=/35− | 3+/44=/33− | 6+/25=/49− | 6+/18=/56− |
| 41 | 1+/44=/35− | 2+/40=/38− | 6+/27=/47− | 6+/18=/56− |
| 42 | 11+/53=/16− | 20+/46=/14− | 9+/27=/44− | 8+/19=/53− |
| 43 | 0+/57=/23− | 4+/48=/28− | 10+/22=/48− | 6+/19=/55− |
| 44 | 8+/56=/16− | 16+/46=/18− | 12+/24=/44− | 8+/20=/52− |
| 45 | 5+/56=/19− | 15+/49=/16− | 9+/27=/44− | 7+/20=/53− |
| 46 | 1+/47=/32− | 3+/41=/36− | 7+/25=/48− | 8+/17=/55− |
| 47 | 0+/41=/39− | 4+/37=/39− | 5+/27=/48− | 6+/19=/55− |
| 48 | 2+/41=/37− | 1+/36=/43− | 6+/26=/48− | 6+/17=/57− |
| 49 | 12+/54=/14− | 22+/42=/16− | 13+/24=/43− | 8+/21=/51− |
| 50 | 2+/47=/31− | 3+/42=/35− | 8+/24=/48− | 7+/17=/56− |

**Table 4.** Comparison of HA-NEAT-DBM heuristics on different benchmarks of CEC 2021, Mann–Whitney tests, run 49.

| Run | NL-SHADE-RSP$_s$ | |
|---|---|---|
| | 10*D* | 20*D* |
| Basic (000) | 4+/6=/0− | 2+/7=/1− |
| Bias (100) | 2+/7=/1− | 3+/6=/1− |
| Shift (010) | 1+/6=/3− | 3+/4=/3− |
| Rotation (001) | 2+/7=/1− | 4+/5=/1− |
| Bias, Shift (110) | 1+/5=/4− | 4+/3=/3− |
| Bias, Rotation (101) | 1+/8=/1− | 4+/5=/1− |
| Shift, Rotation (011) | 0+/9=/1− | 1+/6=/3− |
| Bias, Shift, Rotation (111) | 1+/6=/3− | 1+/6=/3− |

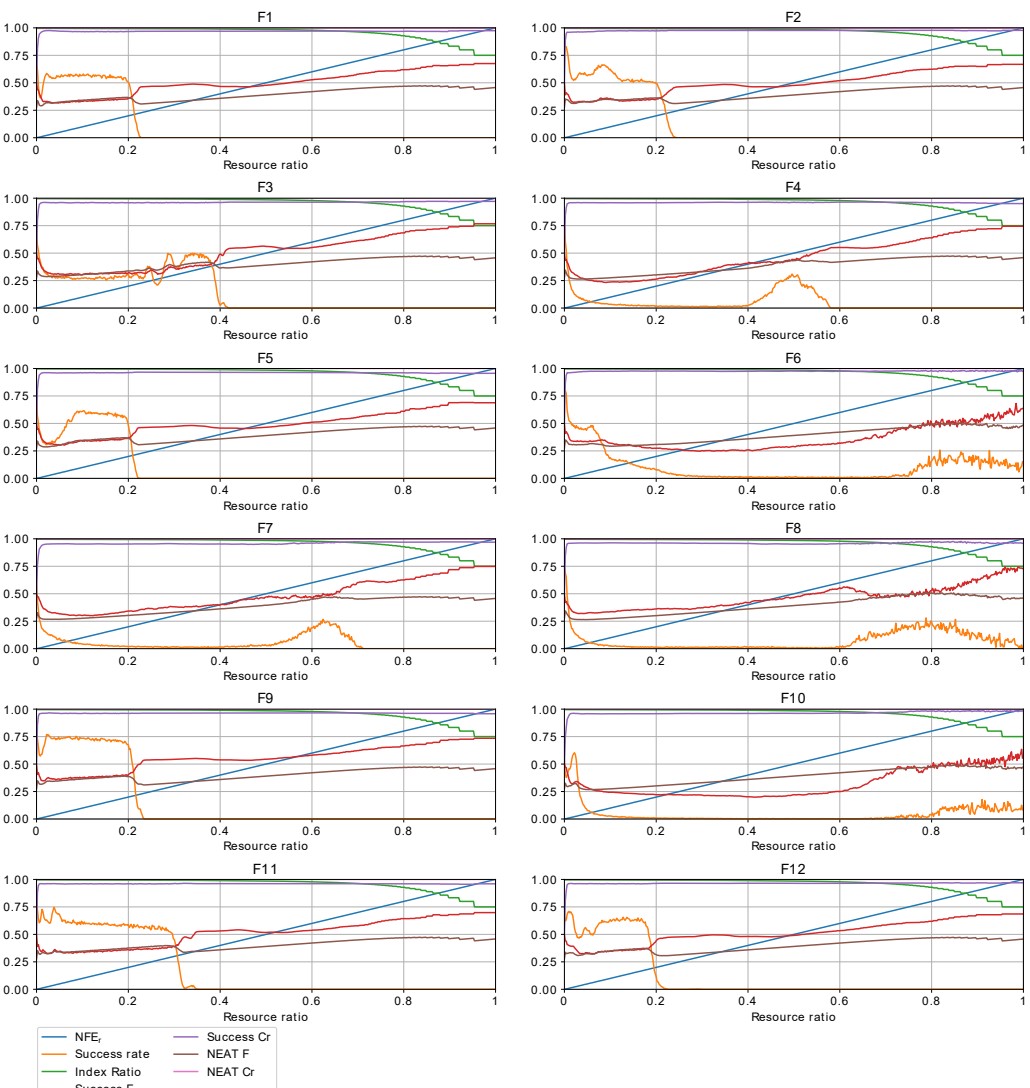

**Figure 1.** Process of parameter adaptation with solution from run 29, CEC 2022 test functions, 10*D*.

The curves shown in Figure 1 represent the inputs to the NEAT solution, as shown in Section 2.3 of this paper, and the outputs, which will be used to sample *F* and *Cr* with Cauchy and normal distribution. Here, the Success *F* and Success *Cr* are the last successful parameter values, which are averaged over the whole population, and the index ratio is shown only for the last individual. The resulting *F* and *Cr* values are also averaged as they depend on the individual index ratio. Analyzing the behavior of parameter adaptation, it can be noted that the resulting HA-NEAT-DBM *F* are highly dependent on the successful *F*

values; in particular, they are following opposite trends: If a successful *F* increases, then the resulting *F* decreases. For example, after around 20% of the computational resource on F1, F2, F5, F9, F11 and F12. Another observed dependence is on the success ratio: If it increases, then the resulting *F* values also increase, and vice versa. In addition, the index of the individual has a certain influence: It can be observed by the step-like curve of the resulting *F*, which is similar to the index ratio. The index ratio of the last individual has these steps at the end of the search because the population size decreases and the last individual at the end has an index of 3 (out of 4), giving a value of 0.75. In general, the resulting *F* follows the successful values, sometimes being below or above them. As for the *Cr* values, the solution from run 29 seems to set them as high as possible, and for all functions the resulting *Cr* values were set to 1. This may explain the high performance on rotated functions.

Visualizing the considered solution from run 29 could be helpful in understanding the reasons for such behavior. Figure 2 shows the NEAT solution graph for run 29.

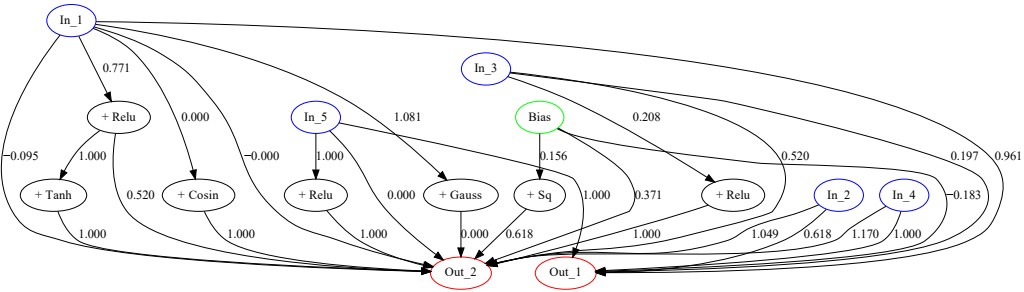

**Figure 2.** Graph of solution from run 29.

Figure 2 shows that, for example, input 4 (successful *F*) has a direct influence on output 1 (resulting *F*) with a weight coefficient of 1. This means that there is a direct influence. However, input 3 (individual index) also has a positive coefficient. That is, the larger the index of the individual in a sorted array, the larger the *F* values set for it. Thus, worse individuals receive higher *F* values, while the best ones have smaller *F* values. Such a mechanism is similar to the sorting of crossover rates used in NL-SHADE-RSP. As for the crossover rates, inputs to *Out*_2 are mostly positive, resulting in very high values, which are then truncated to the [0, 1] range.

Figure 3 shows the graph of the solution from run 49, and Figure 4 shows the process of parameter adaptation for the CEC 2021 benchmark. Solution from run 49 is chosen for comparison as it has shown the most promising results.

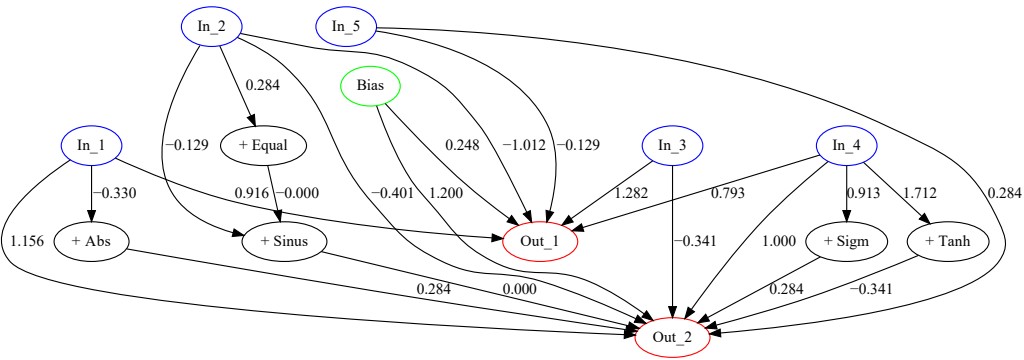

**Figure 3.** Graph of solution from run 49.

The parameter adaptation technique from run 49 is not very complicated and relatively easy to analyze. For example, the successful *F* values as well as the individual index and current computational resource ratio have a direct positive influence on the resulting *F*. As for the crossover rate, again, the successful *Cr* values have a weight value of 0.284, meaning

that these values are not as important as some others. However, successful *F* values have an influence on the resulting *Cr*, in particular, there are three paths from input 4 to output 2 with sigmoid and tanh functions. The graphs in Figure 4 show how this solution works on biased, shifted and rotated benchmark functions.

As can be seen from Figure 4, here, the resulting crossover rates *Cr* have more complex behavior. They are dependent on other variables and, in general, are reducing closer to the end of the search process. The resulting *F* values mainly follow the successful *F* values, but there is an influence of the success rate. In general, however, *F* values are increasing closer to the end of the search.

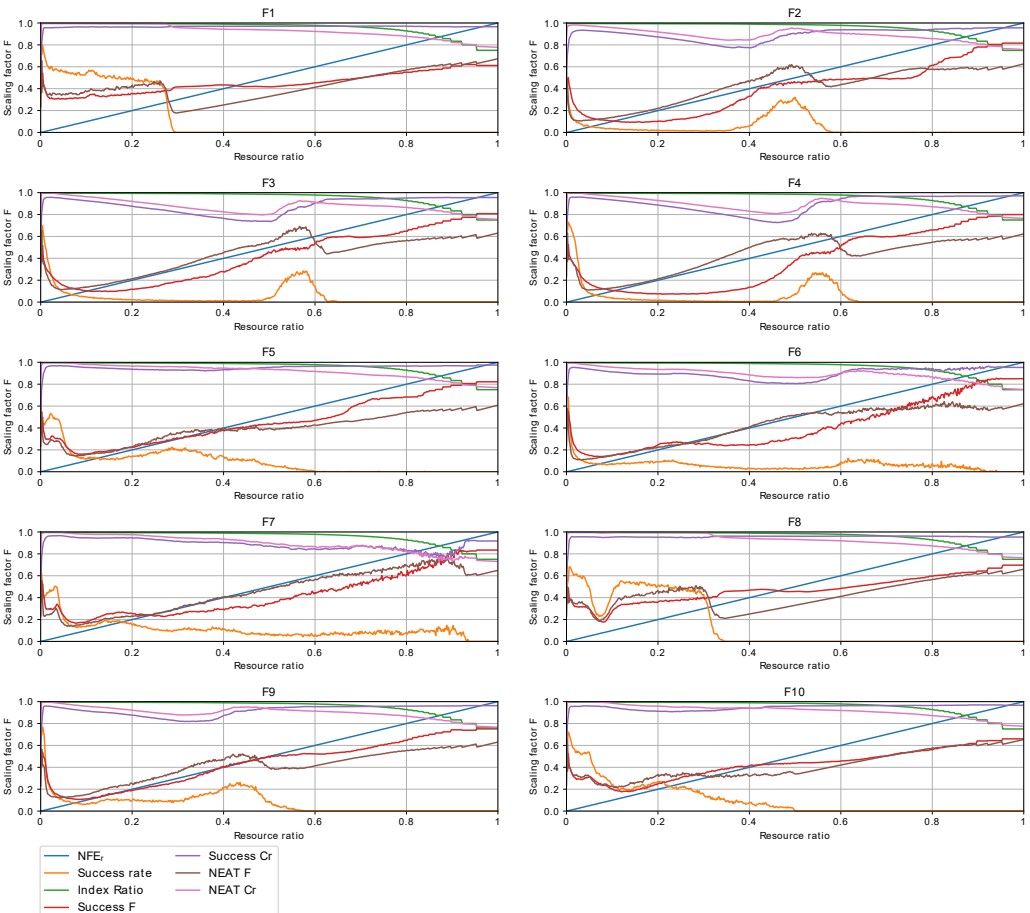

**Figure 4.** Process of parameter adaptation with solution from run 49, CEC 2021 test functions, 10*D*, biased, rotated and shifted functions.

In the next section, the discussion of the presented results is given.

## 4. Discussion

The results of the computational experiments presented in the previous section demonstrate the possibility of applying a neuroevolutionary approach to the problem of the automatic design of parameter adaptation techniques. The results also show that there is a certain overfitting to the problems which are used for training. Thus, to receive more generalized solutions capable of controlling the parameters of DE efficiently, more training cases should be considered. Moreover, these training scenarios should be diverse, i.e., represent different cases which may occur during the search by DE. In addition, the dependence on the available computational resource should be analyzed: In the performed experiments, the computational resource during the whole training process was fixed, and the testing was performed with the same resource.

The approach used in this study follows the ideas of automatic heuristic generation presented in several studies, such as [64–67]. These hyper-heuristics are aimed at designing solutions for complex problems with heuristic approaches, and one of our previous studies has shown that it is possible to use genetic programming for the problem of designing parameter adaptation techniques [22]. The hyper-heuristics are important for several reasons: They not only create a solution to the problem (which could be in the form of an algorithm, graph or neural network), which could be directly used, but also discover new, non-trivial ways of solving problems. This is possible due to the fact that the algorithms are random and are not biased by the human perception of the problem. As an example, to the best of our knowledge, there are no efficient parameter adaptation techniques for DE, which would use success ratio values directly, and here, it appeared to be informative for generating $F$ and $Cr$. Moreover, such values as the ratio of an index of an individual in the population sorted by fitness is an example of an input value, which is quite difficult to apply for parameter adaptation, simply because its influence is unclear. However, the NEAT approach, for example, HA-NEAT-DBM used in this study, is able to make use of these values and apply them to the samples $F$ and $Cr$. Of course, this does not mean that these values are reliable information sources; however, the analysis of solutions may help researchers gain insights into how these values could be of use and what the internal mechanisms could be blocking the algorithm from achieving better search results.

In this manner, applying hyper-heuristic approaches to parameter adaptation in DE could be considered as a knowledge extraction procedure, or even 'algorithm mining' (an analogy for data mining). An important part of this is that the computational experiments are performed without a high-level theory, which would describe the possible outcomes and predict the results. Here, the experiment has 'a life of its own', where massive computation and application of evolutionary principles lead to automatic knowledge extraction. Such studies are known to follow the concept of new experimentalism [68], which is a part of any experimental research in the area of evolutionary computation.

Taking into account all of the above, the application of hyper-heuristics to parameter adaptation is a research direction where new and important results could be found for the whole evolutionary algorithms area, and the particular learning algorithm used is not so important: should it be genetic programming, neuroevolution, fuzzy logic system or something else, while it allows the direct analysis of solutions, it can be considered as a knowledge extraction method.

## 5. Conclusions

In this study, the modification of neuroevolution of augmented topologies, in particular, the HA-NEAT-DBM algorithm with heterogeneous activation functions, difference-based mutation, lexicase selection and behavior-based speciation, was applied to the problem of designing parameter adaptation techniques. The performed computational experiments have shown that the proposed approach is able to create efficient adaptation schemes, which behave differently from existing well-known mechanisms, such as success-history adaptation. The analysis of the best solutions has shown that the designed heuristics are relatively simple, although they use other information sources compared to existing state-of-the-art approaches. Further directions of study in the area of generative hyper-heuristics for parameter adaptation may include training on larger sets of benchmark functions, applying them to population size adaptation, and using other mechanisms than neuroevolution.

**Author Contributions:** Conceptualization, V.S. and S.A.; methodology, V.S., S.A. and E.S.; software, V.S. and E.S.; validation, V.S., S.A. and E.S.; formal analysis, S.A.; investigation, V.S.; resources, E.S. and V.S.; data curation, E.S.; writing—original draft preparation, V.S. and S.A.; writing—review and editing, V.S.; visualization, S.A. and V.S.; supervision, E.S.; project administration, E.S. funding acquisition, S.A. and V.S. All authors have read and agreed to the published version of the manuscript.

**Funding:** This work was funded by the Ministry of Science and Higher Education of the Russian Federation, State Contract FEFE-2020-0013.

**Institutional Review Board Statement:** Not applicable.

**Informed Consent Statement:** Not applicable.

**Data Availability Statement:** Not applicable.

**Conflicts of Interest:** The authors declare no conflict of interest.

## Abbreviations

The following abbreviations are used in this manuscript:

| | |
|---|---|
| CI | Computational Intelligence |
| ANN | Artificial Neural Networks |
| FLS | Fuzzy Logic Systems |
| EA | Evolutionary Algorithms |
| DE | Differential Evolution |
| HH | Hyper-Heuristic |
| ADA | Automated Design of Algorithms |
| NEAT | Neuroevolution of Augmented Topologies |
| CEC | Congress on Evolutionary Computation |
| GP | Genetic Programming |
| BCHM | Bound Constraint Handling Method |
| SHADE | Success-History Adaptive Differential Evolution |
| LPSR | Linear Population Size Reduction |
| NLPSR | Non-Linear Population Size Reduction |
| DBM | Difference-Based Mutation |

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
