# Peer review of "Neuroevolution for Parameter Adaptation in Differential Evolution"

_algorithms, doi:10.3390/a15040122_

Round 1

Reviewer 1 Report

The paper deals with parameter adaption in differential evolution (DE). Therefore, a adaptive neural network of augmented topologies is employed.  The paper discusses the proposed scheme and presents numerical experiments showing its effectiveness. The results are convincing. In general, I think this is a good paper which fits the scope of the journal. However, an additional check of grammar and style is needed, for instance the spelling in the captions of Fig. 2 and 3 should be corrected.

Author Response

Thank you for the a good evaluation of our work. We have performed checking the grammar and spelling with a native speaker in the whole manuscript.

Reviewer 2 Report

This paper describes the use of neuroevolution, namely the NEAT algorithm, to evolve strategies for parameter adaptation in differential evolution. The article is well written, with only minor issues easy to correct in a final editing. The introduction makes a good job of presenting the field and this particular problem to the readers. The Materials and Methods section thoroughly describes not only the differential evolution algorithm, but also the NEAT algorithm used to evolve the neural networks that represent the parameter adaptation strategies. It also describes in detail the approach proposed by the authors, referred to  as HA-NEAT-DBM (Heterogeneous Activation NEAT with Difference-Based Mutation).

The algorithm was tested using the CEC 2022 benchmark functions, facilitating its comparison with other contemporaneous approaches. The experimental setup is well described, thus allowing for the replication of these experiments. The designed parameter adaptation heuristics where compared with other state-of-the-art approaches using adequate statistical tools. The results obtained are thoroughly discussed and seem to support the authors’ conclusions.

Overall I found the article an interesting read, the proposed approach, original, and the methodology used, sound. My only criticism is that such a computational intensive approach may not be viable in many problems or even competitive with simple approaches such as grid searching for good parameters. Even so, the information obtained with these experiments can always be useful for designing or guiding other approaches. 

Author Response

(The authors gave the same response as above.)
